# PPChain: A Blockchain for Pandemic Prevention and Control Assisted by Federated Learning

**DOI:** 10.3390/bioengineering10080965

**Published:** 2023-08-15

**Authors:** Tianruo Cao, Yongqi Pan, Honghui Chen, Jianming Zheng, Tao Hu

**Affiliations:** Science and Technology on Information Systems Engineering Laboratory, National University of Defense Technology, Changsha 410073, China

**Keywords:** blockchain, federated learning, pandemic prevention and control, privacy-preserving

## Abstract

Taking COVID-19 as an example, we know that a pandemic can have a huge impact on normal human life and the economy. Meanwhile, the population flow between countries and regions is the main factor affecting the changes in a pandemic, which is determined by the airline network. Therefore, realizing the overall control of airports is an effective way to control a pandemic. However, this is restricted by the differences in prevention and control policies in different areas and privacy issues, such as how a patient’s personal data from a medical center cannot be effectively combined with their passenger personal data. This prevents more precise airport control decisions from being made. To address this, this paper designed a novel data-sharing framework (i.e., PPChain) based on blockchain and federated learning. The experiment uses a CPU i7-12800HX and uses Docker to simulate multiple virtual nodes. The model is deployed to run on an NVIDIA GeForce GTX 3090Ti GPU. The experiment shows that the relationship between a pandemic and aircraft transport can be effectively explored by PPChain without sharing raw data. This approach does not require centralized trust and improves the security of the sharing process. The scheme can help formulate more scientific and rational prevention and control policies for the control of airports. Additionally, it can use aerial data to predict pandemics more accurately.

## 1. Introduction

The outbreak of coronavirus disease 2019 (COVID-19) had a huge impact on the world economy and people’s lives. It is in the interest of all mankind to contain a pandemic at an early date. However, the spread of the pandemic is affected by many factors. Pandemic prevention policies and pandemic prevention psychology in different countries will have different effects [1], thus having a significant impact on the model parameters of virus transmission. Raffetti et al. [2] demonstrated that national policies are the most important factor affecting the spread of the pandemic. Sweden adopted a “natural” herd-immunity strategy to deal with the pandemic, but this model resulted in a COVID-19 death rate 10 times higher in Sweden than that in neighboring Norway [3]. Mishra et al. [4] compared Denmark, Britain, and Sweden under different policy models. They concluded that domestic policy effects are affected by inter-country population flows and that similar policies may even produce different effects.

Therefore, it is necessary to include population flow between countries and regions when designing strategies for the prevention and control of a pandemic, as well as develop effective policies about population flow to contain the outbreak. This paper aims to study the impact of air network transmission on the pandemic and how to formulate shipping policies to contain the pandemic. However, at present, most national medical centers are often unable to know the flight status of confirmed patients, and most airline companies cannot obtain the illness status of passengers due to different national policies. Meanwhile, federated learning [5] can realize multi-party data integration without sharing data. In this paper, federated learning is chosen instead of other distributed learning frameworks.

The main reason is that the compute nodes have absolute control over the data in federated learning, and the central server cannot directly or indirectly operate the data of the compute nodes. The compute nodes can stop computing and communication at any time and exit the learning process. However, in other distributed machine-learning frameworks (such as MapReduce, etc.), the central server has a high level of control over the compute nodes and their data. The compute nodes are completely controlled by the central server and receive instructions from the central server. For instance, MapReduce’s central server can issue instructions to the compute nodes to exchange data with each other. This can potentially compromise the privacy of user data and may add additional communication overhead. Meanwhile, the raw data of federated learning can be kept locally, which is an advantage over other distributed machine-learning approaches.

The traditional federated learning model often requires an aggregator. This will lead to privacy and model failure problems [6] if the center aggregator is attacked. Our work is dedicated to enabling federated learning in association with blockchain to engage safely with collaborative training without a central aggregator and apply it to pandemic prevention.

In this paper, we present the following: First, we design a pandemic prevention and control model based on a blockchain. The characteristics of blockchain such as decentralization and security ensure the reliability and effectiveness of the learning process. This causes federated learning to not depend on the third-party central server, but rather depend on the consensus mechanism for better data protection and security. The scheme implements the integration of pandemic data and airline data through federated learning. Second, the impact of aviation network transmission on pandemic prevention and control is obtained, thus providing support for scientific predictions of pandemic changes and auxiliary policy making.

The rest of this article is organized as follows: Section 2 investigates the current situation related to the use of federated learning for pandemic prevention and control. Section 3 mainly elaborates on the main architecture of the model. Section 4 introduces the experiment and analysis. Section 5 summarizes the main work of this paper.

## 2. Related Work and Prior Knowledge

### 2.1. Application of Federated Learning in Epidemic

Qian et al. [7] described real-world cases of using federated learning in COVID-19 as well as non-COVID-19 scenarios and analyzed its limitations and practical challenges. References [8,9,10,11,12,13] used federated learning to assist in the diagnosis and intelligent monitoring of COVID-19. However, the above schemes only improved the accuracy of the diagnosis of suspected patients but did not discuss how to evaluate the effect of the policy. Chen et al. [14] constructed a COVID-19 vulnerability prediction map using federated learning synergy to identify high-risk areas and reduce the spread of the disease. However, the method is mainly oriented toward the collaboration of organizations in the same field and does not provide a cross-domain collaboration approach. Samuel et al. [15] proposed a privacy architecture based on federated learning and blockchain technology to support the cross-domain interaction regarding COVID-19 information and protect the authenticity and privacy of this information. Pang et al. [16] fused urban digital twins between multiple cities through federated learning technology and constructed a collaborative urban crisis management paradigm to explore and formulate effective prevention and control policies. However, the method lacks the competence to analyze the impact of population movement between regions on pandemic prevention.

### 2.2. Blockchain Technology

In 2008, Satoshi Nakamoto published a white paper on Bitcoin [17], marking the birth and landing of blockchain technology. Blockchain technology integrates P2P networks, confidential algorithms, consensus mechanisms, and other technologies that can be used for secure and trusted data storage, transmission, and operation. Today, the blockchain has developed into the “blockchain 3.0” era [18], and it is integrated with various industries through smart contracts [19]. The essence of blockchain technology is a distributed shared ledger. In an environment of incomplete trust, a P2P network is built to verify and store data through a chain data structure, and a consensus mechanism and encryption algorithm are adopted to achieve trusted transmission and operation of data. A smart contract is an automatically executed association deployed on the blockchain based on established rules [20], which is another form of blockchain consensus. Once the condition is met, execution is triggered and cannot be terminated. Smart contracts have evolved from scripts to programming languages. Bitcoin’s scripting support is limited to programming and is not Turing-complete. Ethereum [21] provides users with a Turing-complete programming language but requires users to learn additional specific languages. Hyperledger [22] supports users to write smart contracts in programming languages such as GO, and uses Docker containers as the running environment, lowering the development threshold for writing smart contracts. Basetty Mallikarjuna et al. [23] applied deep neural network (DNN) analysis in healthcare and the COVID-19 pandemic and presented smart contract procedures to identify feature data (FED) extracted from existing data. Shan Jiang et al. proposed BlocHIE [24], a Blockchain-based platform for healthcare information exchange. They also designed a bloom filter [25] to select a low-frequency keyword from the multiple keywords input by the ITS data owner.

### 2.3. Application of Blockchain in Federated Learning

Chen et al. [26] analyzed the privacy and security issues of the learning model and designed a federated learning system that supports privacy protection based on blockchain, replacing the central server with parameter aggregation. Ramanan et al. [27] proposed a federated learning environment based on blockchain which uses blockchain to store and share global models and perform model aggregation tasks through smart contracts. Rehman et al. [28] proposed a set of blockchain-based federated learning frameworks for mobile edge computing networks, redefining the model’s storage mode, training process, and consensus mechanism. Anik Islam et al. [29] presented a federated learning-based data-accumulation scheme that combines drones and blockchain. Zhu et al. [30] comprehensively surveyed challenges, solutions, and future directions for blockchain-empowered federated learning.

To summarize, federated learning for pandemic analysis is currently mainly used to integrate knowledge in the field and relies on blockchain technology to strengthen the robustness and credibility of federal learning. However, the research on the interaction between regional policies is still insufficient. On the other hand, the storage overhead of a blockchain is large, which has a great impact on efficiency when federated learning requires more model parameters. Therefore, it is necessary to study how to integrate cross-domain information while protecting the rights and interests of data owners. Through this method, we aim to identify the impact of inter-regional population flow on pandemic prevention and control to better develop pandemic predictions and policy making.

## 3. PPChain

In this section, we propose a blockchain for COVID-19 prevention and control based on federated learning. With blockchain technology, data can be trusted and distributed among multiple parties without a third-party central authority, reducing the risk of data leakage. There are two main types of participants in the network: regional pandemic management centers and individual airlines. Using the federated learning technique, the two groups of entities can figure out how the disease is spreading through flights without sharing raw data, which allows for the development of targeted containment policies and predictions. The system uses smart contracts to act as aggregators. In addition, to ensure the efficiency of the blockchain network, the models are trained locally and the blockchain only utilizes parameter information flow. The system architecture is shown in Figure 1.

According to the above analysis, the pandemic mainly spreads and spreads rapidly through the transportation network. Therefore, we choose the representative transportation route with the widest spread and fastest speed, namely the flight, to study the transmission mode of the pandemic, to better predict and prevent the pandemic. This can also be extended to multiple transport networks. The transmission of the pandemic should be coupled with the transportation network, but the traditional pandemic analysis and prediction models such as SIR Do not consider the factors of the transportation network. However, the current research on the effect of transportation networks on pandemic transmission does not have the support of pandemic transmission dynamics. Therefore, this paper hopes to train the SIR model in collaboration with the communication model of the traffic network, to better predict and prevent the pandemic. Specifically, federated learning enables collaborative learning of two different domain models without sharing their respective training data. In addition, the effect of traffic network changes can be directly transmitted to the change in epidemic model parameters without lag. Blockchain provides the building blocks for trusted collaboration and protects the security of training. The traditional combination of blockchain and federated learning is mostly collaborative training between the same models, and this paper is inclined to study collaborative training between different models.

Using the framework shown in Figure 1, the training process of federated learning is divided into two parts. First, the sample space is aligned based on people with the same identity information who are distributed to different parties. Using the user ID alignment technique based on encryption ensures that the original data of each party do not need to be exposed. In the second phase, the encryption model training is performed based on these aligned entities as follows:After identifying the shared entity, the blockchain triggers the smart contract to create the key pair and send the public key to regional pandemic management centers and individual airlines.Regional pandemic management centers and individual airlines encrypt and exchange intermediate results, which are used to help calculate gradients and loss values.Regional pandemic management centers and individual airlines calculate the encryption gradient and add additional masks, respectively. Regional pandemic management centers also calculate the encryption loss. The two parties then write the encrypted results into the blockchain’s ledger through the SDK.Smart contracts on the blockchain decrypt the gradients and losses newly written into the ledger and send the results back to both parties. Regional pandemic management centers and individual airlines unmask the gradient information and update the model parameters according to the gradient information.Meanwhile, the parameters trained by the model are written into the blockchain. Then, the parameters can be read for better pandemic prediction and prevention and can help control policy formulation.

Through the above process, the spread of infection caused by the flight flow between different regions can be calculated. In this study, we used the SIR model (susceptible–infected–removed) as the model to be trained.

### 3.1. Pandemic Transmission Model Based on SIR

The SIR model mainly simulates the evolution of susceptible population *S*, infected population *I*, and removed population *R* (including those with immunity, those who are no longer infected, or those who have died after being cured). The key parameters of the model are shown in Table 1, where effective reproduction number *R* is the core parameter trained by federated learning.

The SIR model is calculated as follows:(1)$$\frac{dS}{dt}=-\frac{\beta }{N}IS$$
(2)$$\frac{dS}{dt}=\frac{\beta }{N}IS-\gamma I$$
(3)$$\frac{dR}{dt}=\gamma I$$
where *t* is time, S(t) is the number of susceptible persons at *t*, I(t) is the number of infected persons at *t* and R(t) is the number of cured persons at *t*.
(4)N=S+I+R

In the initial state, $S\left(0\right)=S_{0}$, $I\left(0\right)=I_{0}$ and $R\left(0\right)=R_{0}$. According to Formulas (1) and (3): (5)$$S=S_{0}e^{\left[\frac{\beta }{N\gamma }\left(R-R_{0}\right)\right]}$$

Eventually, I(t) approaches zero according to Formulas (4) and (5); thus, it can be solved with: (6)$$R_{all}=N-S_{0}e^{\left[\frac{\beta }{N\gamma }\left(R_{all}-R_{0}\right)\right]}$$

We accumulate the time series C(t) that is formed by the daily data, as shown in Formula (7).
(7)$$C\left(t\right)=I\left(t\right)+R\left(t\right)=N-S_{0}e^{\left[\frac{\beta }{N\gamma }\left(R-R_{0}\right)\right]}$$

Therefore, the least squares method is used to conduct regression analysis on the time series formed by the confirmed data and obtain the estimation of each parameter, as shown in Formula (8).
(8)$$\underset{\beta ,\gamma }{argmin}\left(\right\|C_{t}-\hat{C_{t}}\left(\beta ,\gamma ,S_{0}\right)\left\|\right)$$

To protect the patients’ private information in the electronic medical record data, we adopted the federal learning method to fine-tune the model training. Formula (8) is the objective function of federated learning.

### 3.2. Aviation Network Transmission Model

To determine the influence of air traffic network propagation, the probability of cities being affected by diffusion is calculated according to the parameters transmitted by federated learning. The susceptibility probability of flights taking off from each city is approximated as follows: (9)$$P_{plane}=R_{plane}\cdot \frac{I_{plane}}{N_{plane}}$$

$R_{plane}$ is the effective reproduction number of the flight, $I_{plane}$ represents the presence of cases on the flight and $N_{plane}$ is the total number of people on the flight. Thus, the probability of city *k* being affected by the imported spread of the pandemic can be calculated, as shown in Formula (10).
(10)$$P_{k}=1-P_{k}=1-\prod_{n=1}^{num_{k}}\left[\prod_{m=1}^{f_{n}}\left(1-P_{m}\right)\right]$$

$num_{k}$ represents the number of cities which have flights to city *k*, $f_{n}$ represents the number of flights from city *n* and $P_{m}$ is the susceptibility probability of flight *m*.

Based on the probability of city *k* being affected, the importance of airports is ranked by measuring the location of airport nodes and the airline flow of airports. A weighted proximity algorithm is used to highlight the influence of path distance between nodes on recognition results.
(11)$$P_{i}^{C}=P_{k}\cdot \left(M-1/\sum_{j=1}^{M}f\left(d_{ij}\right)\right)$$

*M* represents the number of airports in the network, and $d_{ij}$ represents the minimum number of transit times from airport *i* to airport *j*, where f() is a logarithmic function. The importance output matrix of adjacent nodes is set for evaluation:(12)$$H=\left[\begin{array}{cccc} 1 & \delta_{12}D_{2}/k^{2} & \cdots & \delta_{1M}D_{M}/k^{2}\\ \delta_{12}D_{2}/k^{2} & 1 & \cdots & \delta_{2M}D_{M}/k^{2}\\ \vdots & \vdots & \cdots & \vdots \\ \delta_{M1}D_{1}/k^{2} & \delta_{M2}D_{2}/k^{2} & \cdots & 1 \end{array}\right]$$

In the matrix, $\delta_{ij}$ is the contribution allocation parameter. If two nodes are connected, the value is 1; otherwise, the value is 0. The element on the diagonal is 1, which means that the contribution ratio of the node to itself is 1. $D_{i}$ is the degree of the airport *i* and *k* is the average degree of the node: (13)$$k=\sum_{i=1}^{M}D_{i}/M$$

Then, the weight $S_{i}$ of each airport node is calculated: (14)$$S_{i}=\sum_{j\in N_{i}}W_{ij}$$

$N_{i}$ is the neighbor node set of node *i* and $W_{ij}$ is the weight of the edge directly connected to node *i*. According to Formulas (11)–(14), the importance of each airport node is obtained as $C_{i}$.
(15)$$C_{i}=\sum_{i=1,j\neq i}^{M}P_{j}^{C}\delta_{ij}S_{j}D_{j}/k^{2}$$

### 3.3. Federated Learning Training Process Assisted by Blockchain

The workflow of PPChain mainly includes four stages: initializing the collaborative training alliance, writing the encrypted results to the ledger, reading and sending the results, and updating the model parameters.

In the stage of initializing the collaborative training alliance, let us assume that the participants of $N_{1}$ (CDC (Centers For Disease Control And Prevention) $p_{i},i\in \left\{1,2,\ldots ,N_{1}\right\}$), and the participants of $N_{2}$ (airlines $q_{i},i\in \left\{1,2,\ldots ,N_{2}\right\}$) join the blockchain network and obtain a configuration file containing predefined information such as a collaboration model and initialization parameters to form a collaborative training alliance. The blockchain network randomly selects $M(M=M_{1}+M_{2},\;\mathrm{where}\;M_{1}=N_{1}/2,M_{2}=N_{2}/2)$ participants to form a certification consortium to enable the system’s consensus algorithm.

It is a remarkable fact that smart contracts will be deployed on the blockchain to create a key pair for each entity that joins the network and will return the public key as a means of identification and encryption, while the private key will be stored in the blockchain ledger along with the entity’s characteristic information. It will be read by the smart contract for decryption during the result reading and sending phase.

To ensure communication efficiency and make collaborative training and node identity information easy to maintain, the channel mechanism is applied to the blockchain network. The channel is a dedicated subnet for specific members to communicate with each other. Different types of transactions will be executed in different task channels. Therefore, a relatively independent ledger will be formed for easy management and maintenance, and finally, communication efficiency will be improved.

In this paper, two types of channels were designed in which the identity channel is used for the storage of private keys and entity characteristic information. We designed an identity form for the registration of information for nodes joining the network. Forms consist of normalized data containing entity identity information that circulates in channels in the form of smart contracts. The typical data structure is shown in Table 2.

The other channel is used for the collaborative training of the federated learning method, where relevant entities join and complete business on the channel after authentication and authorization. Additionally, the channel ledger records the complete process of collaborative training. When the training is completed, the ledger data of the channel are hashed to form a hash value and stored in the system’s general ledger. Similarly, collaborative training is also implemented via forms in the channel through smart contracts. Table 3 shows the form structure of the co-training process.

In the stage of writing the encryption results to the ledger, the entity that uploads the model fills in the corresponding form information and iteratively updates the epoch field. The form is broadcast on the channel. The certification consortium verifies and signs off on transactions. The transactions are handed over to channel members for consensus. If a consensus is reached, the transaction-committing channel is packaged into transaction blocks.

After the above steps, the smart contracts on the blockchain decrypt the gradients and losses newly written to the ledger and send the results back to both parties, which are the results of the reading and sending phase. The pandemic control centers and airlines will uncover the gradient information. In the model parameter update stage, the model parameters are updated according to the gradient information, and the above process can be summarized as the collaborative training step in Table 4.

## 4. Experiment and Analysis

This section introduces the experimental environment of PPChain and evaluates the system performance from four aspects: training accuracy, time cost, transaction processing efficiency, and transaction latency.

Hyperledger Fabric was used to build the blockchain network in the system, and the federated learning parameters were transmitted through smart contracts. The main body of the system model was realized under the chain, and SDK was used to read and write the data on the chain.

### 4.1. Experiment Settings

To evaluate the system performance of the proposed PPChain, experiments were run on real datasets. Global pandemic data published by John Hopkins University were used as the pandemic data, and public datasets of flight connections were used as the inter-regional flight information.

In this experiment, the two types of datasets were randomly divided into 10 subsets with equal numbers and assigned 20 participants to create local datasets.

The experiment was run on a laptop equipped with a 4-core, 8-thread Intel CPU i7-12800HX and 32 GB of memory. The Python programming language was used to develop SDKs to implement the business logic of the system. Smart contracts were written in the Go language. The learning model was written using Python 3.9 and PyTorch 1.4.0 and executed on the NVIDIA GeForce GTX 3090Ti GPU.

### 4.2. Performance Evaluation

Table 5 describes the parameters for co-training.

After collaborative training, the training-obtained $R_{0}$ is fitted with the actual infection curve, and the accuracy of the training is judged by RSME, as shown in Figure 2. It can be seen that the system can better implement collaborative training.

In this paper, a more accurate impact of the aviation network on the spread of the pandemic is obtained through collaborative training because this method realizes the data alignment between common entities without exposing privacy. Based on the $R_{plane}$, we applied it to the designed aviation network propagation model to obtain a directed network diagram of the pandemic’s propagation through the aviation network, as shown in Figure 3.

This paper explores the impact of the aviation network on the COVID-19 pandemic to assist in the formulation of effective prevention and control policies. In this paper, the top 20 airports with the highest importance and the top 20 airports with median importance were selected for comparison.

After airport closures, the data can be applied back into the model that has been trained for calculation. By closing these airports, the number of cities affected daily by the imported spread of the pandemic is as shown in Figure 4. From the results, it can be seen that the effect of closing 20 generally important airports is small, but closing the top 20 most important airports will significantly reduce the number of affected cities.

Next, the changes in the transaction efficiency and transaction latency in the collaborative training process of the system were recorded. The transaction processing efficiency is reflected by the system running time, which includes the process of reading data, verifying signatures, completing transactions, achieving consensus, and obtaining data on-chain. Transaction latency mainly includes consensus delay and communication delay between nodes and is divided into maximum delay, average delay, and minimum delay due to network jitter.

In this paper, the number of network nodes was changed to test the changes in transaction efficiency and transaction latency, as shown in Figure 5 and Figure 6.

## 5. Conclusions

In this study, based on the problem that medical centers and airlines cannot fully obtain the flight information of confirmed patients or the illness status of passengers due to privacy protection, we designed PPChain without sharing the original data. The system combines federated learning with a blockchain, improving the security of shared processes without the need for centralized trust. Through experiments, we have verified that effective data-sharing can be achieved without destroying privacy by simulating the impact of aviation policies. Thus, we can formulate prevention and control policies more scientifically and rationally, and predict the spread of a pandemic more accurately.

## Figures and Tables

**Figure 1 bioengineering-10-00965-f001:**
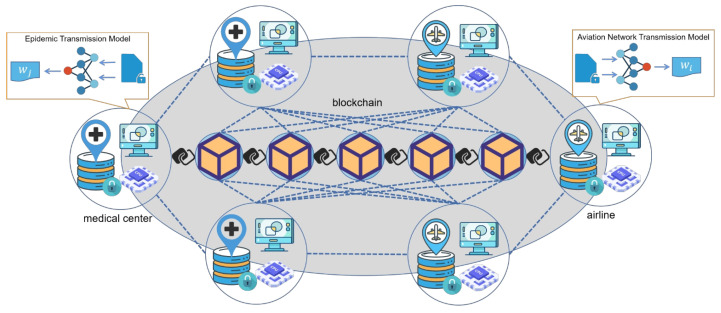
The system architecture.

**Figure 2 bioengineering-10-00965-f002:**
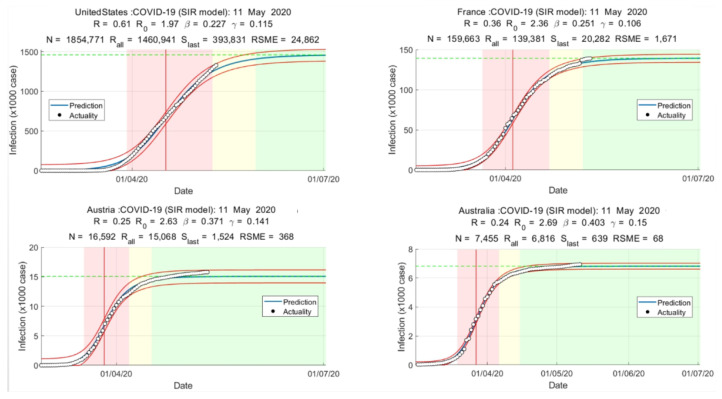
Infection curve fitting graph.

**Figure 3 bioengineering-10-00965-f003:**
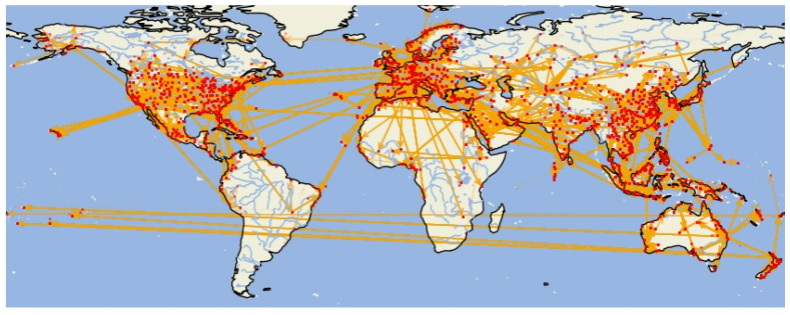
Directed network of pandemic transmission in aviation network.

**Figure 4 bioengineering-10-00965-f004:**
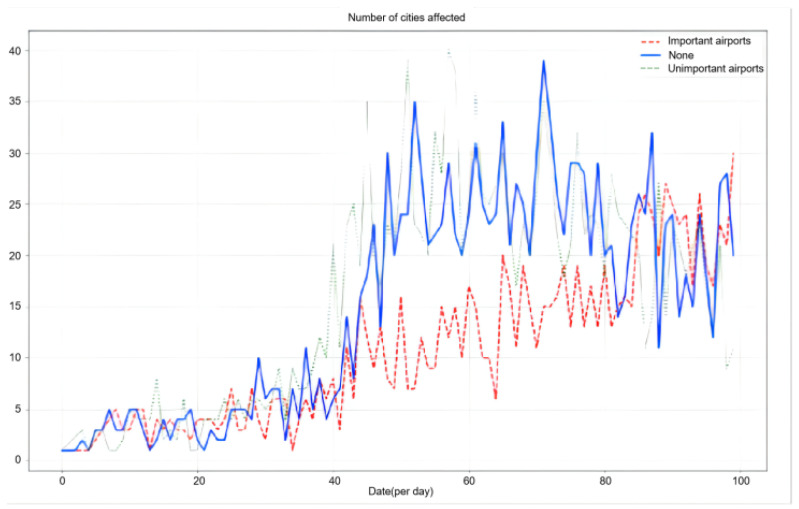
Comparison of the impact of airports on the spread of COVID-19.

**Figure 5 bioengineering-10-00965-f005:**
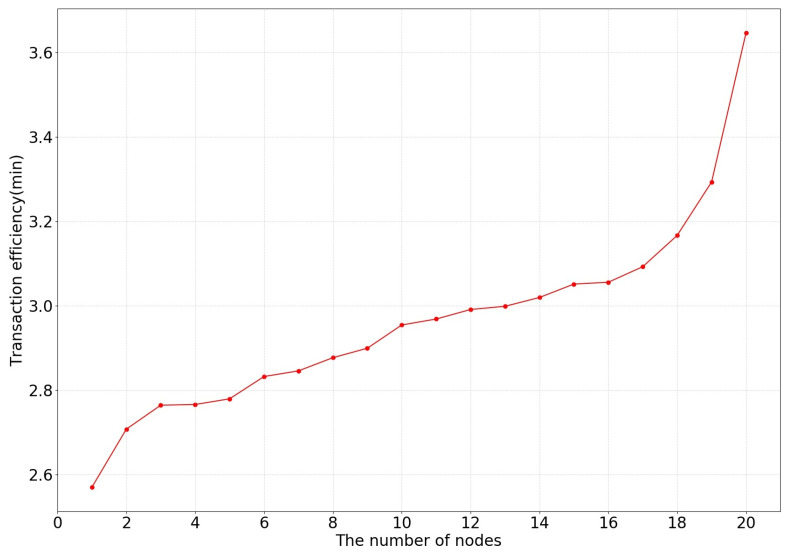
The effect of the number of nodes on transaction efficiency in cooperative training.

**Figure 6 bioengineering-10-00965-f006:**
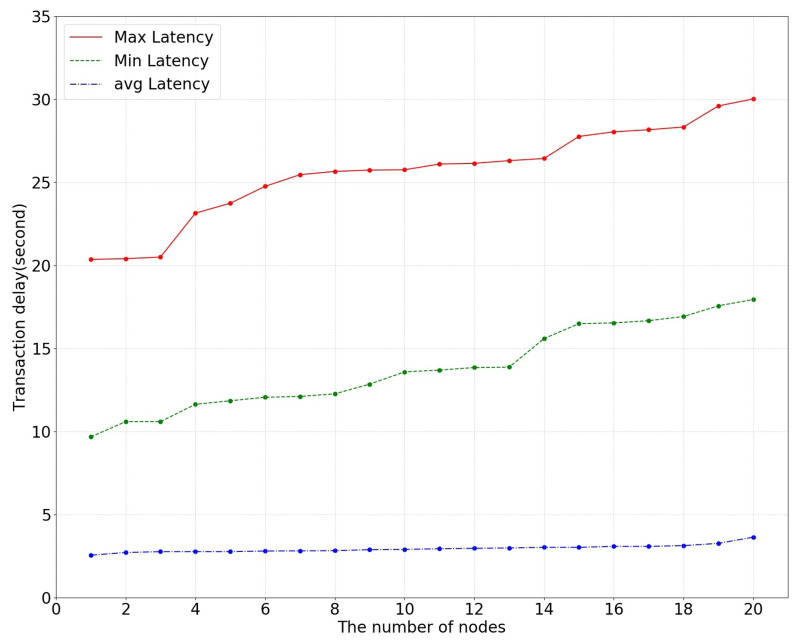
The effect of the number of nodes on transaction delay in cooperative training.

**Table 1 bioengineering-10-00965-t001:** The specifications of SIR model’s symbols.

Symbol	Meaning
*R*	Effective reproduction number (=$\frac{\beta }{\gamma }\left(1-\frac{C}{N}\right)$); the average number of infections per patient
$R_{0}$	Basic reproduction number (=$\frac{\beta }{\gamma }$)
β	Average frequency of exposure (per day)
γ	Average removal frequency (per day)
*N*	Total population (also used as initial susceptible population)
$R_{all}$	Final outbreak size (replaced by final recovery)
$S_{last}$	The rest of the uninfected population
RMSE	Root-mean-square error

**Table 2 bioengineering-10-00965-t002:** The typical data structure of an identity form.

Symbol	Explanation
*ID *	The identity number of the entity
type	Categories of entities (medical, aviation)
name	Name of entity
time	The time that the entity joins the blockchain network
state	Whether the entity is incorporated into an authentication consortium
$s_{k}$	The entity’s private key information

**Table 3 bioengineering-10-00965-t003:** The form structure of the co-training process.

Symbol	Explanation
MissionID	Serial number of the cooperative training task
Owner	The entity that provides the gradient
epoch	The number of rounds iterated by cooperative training
param	Encrypted gradient information
timestamp	The timestamp of the gradient written to the ledger
collaborators	A collection of entities that participate in the corresponding collaboration
mask	Mask information

**Table 4 bioengineering-10-00965-t004:** The proceedings of collaborative training.

Step	CDC	Airport	Smart Contracts on Blockchain
1	Initialize β, γ	Initialize $R_{plane}$	Create an encryption key pair and send the public key to both parties
2	Calculate $\|\Gamma_{A}\left\|=\right\|C_{t}-\hat{C_{t}}\left(\beta ,\gamma ,S_{0}\right)\|$ and send it to the blockchain	Calculate $\|\Gamma_{B}\left\|=\right\|C_{t}-\hat{C_{t}}\left(R_{plane},S_{0}\right)\|$ and send it to the blockchain	Create the collaboration form and write $\left\|\Gamma \|=\right\|\Gamma_{A}\left\|+\right\|\Gamma_{B}\|$
3	Calculate $\frac{\partial \Gamma }{\partial R_{0}}$, then encrypt it and write it to the blockchain	Calculate $\frac{\partial \Gamma }{\partial R_{plane}}$, then encrypt it and write it to the blockchain	Read the contents of the ledger, decrypt them and send to both parties
4	Update β, γ	Update $R_{plane}$	Update block
Obtainedcontent	β, γ	$R_{plane}$	The ledger of co-training results

**Table 5 bioengineering-10-00965-t005:** The parameters for co-training.

Parameter	Value
numberofepochs	10
numberofiterations	1500
learningrate	0.015
batchsize	64

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
