# Peer review of "PPChain: A Blockchain for Pandemic Prevention and Control Assisted by Federated Learning"

_bioengineering, 2023, doi:10.3390/bioengineering10080965_

Round 1
Reviewer 1 Report
The manuscript proposes PPChain, a blockchain-based federated learning approach for pandemic prevention. Although the COVID-19 pandemic is ending, such an approach is potential to help prevent future events. We have comments as follows:
* The main contributions should be summarized at the end of the introduction.
* The manuscript reads like an application of blockchain and federated learning with fewer scientific contributions. The authors should articulate the proposed new approaches.
* In the system architecture, the authors are expected to highlight how federated learning functions. Meanwhile, the authors should justify how different airports and hospitals are motivated to join the system.
* The related work is too brief. Some important work is missing, e.g.,:
- BlocHIE: A Blockchain-based Platform for Healthcare Information Exchange
- Blockchain technology for enhancing supply Chain performance and reducing the threats arising from the COVID-19 pandemic
- Blockchain-empowered federated learning: Challenges, solutions, and future directions
- A blockchain and artificial intelligence-based, patient-centric healthcare system for combating the COVID-19 pandemic: Opportunities and applications
- Privacy-preserving and efficient data sharing for blockchain-based intelligent transportation systems
- Blockchain technology: A DNN token‐based approach in healthcare and COVID‐19 to generate extracted data
The authors are recommended to follow templates to prepare the abstract and introduction.
Author Response
Response to Reviewer 1 Comments
Point 1:
* The main contributions should be summarized at the end of the introduction.
Response 1:
We have clarified our contributions at the end of the introduction.
Point 2:
* The manuscript reads like an application of blockchain and federated learning with fewer scientific contributions. The authors should articulate the proposed new approaches.
Response 2:
The combination of traditional blockchain and federated learning is mostly collaborative training between the same model, and this paper tends to study collaborative training between different models.
Point 3:
* In the system architecture, the authors are expected to highlight how federated learning functions. Meanwhile, the authors should justify how different airports and hospitals are motivated to join the system.
Response 3:
According to the above analysis, the epidemic is mainly spread and rapid spread through transportation networks. Therefore, we chose the most widespread and fastest representative transport route, i.e. flights, to study how the pandemic spreads in order to better predict and prevent it. This can also be extended to multiple transport networks. The spread of a pandemic should be coupled to the transportation network, but traditional pandemic analysis and prediction models such as SIR do not take into account the factors of the transportation network. However, current research on the impact of transportation networks on pandemic transmission dynamics is not supported by pandemic transmission dynamics. Therefore, this paper hopes to co-train the SIR model with the communication model of the transportation network to better predict and prevent the pandemic. Specifically, federated learning enables collaborative learning of two different domain models without sharing their respective training data. In addition, the effects of changes in transportation networks can be passed directly to changes in epidemiological model parameters without lag.
Point 4:
* The related work is too brief. Some important work is missing,
Response 4:
We have reinforced our related works and taken more advanced methods into consideration. With the help of these works we are enabled to illustrate our idea more clearly.
Reviewer 2 Report
The article suggests a solution to the issue of medical centers and airlines not having access to complete flight information or passenger health status due to privacy regulations. They propose using a PPChain that allows for the sharing of information while maintaining the confidentiality of the original data. Essentially, the PPChain encrypts the data through cryptographic methods, enabling information sharing without exposing the underlying data. The idea is interesting. However, I have the following concerns.
1. Please revise the grammatical issues of the paper.
2. Recent blockchain-based FL papers are missing. Some are mentioned as follows.
-> "IoMT: A COVID-19 Healthcare System Driven by Federated Learning and Blockchain," in IEEE Journal of Biomedical and Health Informatics, vol. 27, no. 2, pp. 823-834, Feb. 2023, doi: 10.1109/JBHI.2022.3143576.
-> "FBI: A Federated Learning-Based Blockchain-Embedded Data Accumulation Scheme Using Drones for Internet of Things," in IEEE Wireless Communications Letters, vol. 11, no. 5, pp. 972-976, May 2022, doi: 10.1109/LWC.2022.3151873.
3. Blockchain-based FL is a very popular topic. Please highlight your novelty based on the limitation of existing works.
4. The authors' application context is on Aiport which is missing in the title.
5. Author needs to revise the abstract containing the experimental setup and outcomes.
6. Contribution is not clear
7. Add a summary table for related works (i.e., Section 2) including the limitation of each work.
8. Reference for equations is missing.
9. Authors considered public key encryption. Won't it create an extra delay in the network?
10. Which DL model did the authors consider? How models are aggregated? Blockchain always demands connectivity. Is connection always available? If not then how the model will be collected?
10. Which consensus did the author consider? How data is stored in blockchain? What's the structure of blockchain?
11. Why did the author choose Hyperledger? Won't it be a private network? Who will be the aggregator in the network then?
12. An analysis of computational complexity and convergence is required.
13. Did the authors consider the privacy leakage of the model itself?
14. A Comparison with existing works is missing. Which DL did the author consider?
15. Hight quality figures are required for Fig. 3 and 4.
Please revise the grammatical issues of the paper.
Author Response
Response to Reviewer 2 Comments
Point 1:
Please revise the grammatical issues of the paper.
Response 1:
We have required editor service and revise our grammatical issues.
Point 2:
Recent blockchain-based FL papers are missing
Response 2:
We have reinforced our related works and taken more advanced methods into consideration. With the help of these works we are enabled to illustrate our idea more clearly.
Point 3:
Blockchain-based FL is a very popular topic. Please highlight your novelty based on the limitation of existing works.
Response 3:
The combination of traditional blockchain and federated learning is mostly collaborative training between the same model, and this paper tends to study collaborative training between different models. Specifically, federated learning enables collaborative learning of two different domain models without sharing their respective training data. In addition, the effects of changes in transportation networks can be passed directly to changes in epidemiological model parameters without lag. Blockchain provides the building blocks for trusted collaboration and secures training.
Point 4:
The authors' application context is on Aiport which is missing in the title.
Response 4:
We have mentioned airport in our subtitles, and tried not to limit our application in transportation.
Point 5:
Author needs to revise the abstract containing the experimental setup and outcomes.
Response 5:
Experimental setup and outcomes have been revised as required.
Point 6:
Contribution is not clear
Response 6:
We have clarified our contributions at the end of the introduction.
Point 7:
Add a summary table for related works (i.e., Section 2) including the limitation of each work.
Response 7:
We have summarized the limitation of each work as required.
Point 8:
Reference for equations is missing.
Response 8:
We have updated our reference for equations(SIR)
Point 9:
Authors considered public key encryption. Won't it create an extra delay in the network?
Response 9:
Cryptographic algorithms are not all computed in a blockchain network, and a blockchain network plays more of a role in passing keys. A large number of encryption and decryption operations are done locally.
Point 10:
Which DL model did the authors consider? How models are aggregated? Blockchain always demands connectivity. Is connection always available? If not then how the model will be collected?
Response 10:
There are two models, one is the SIR model and the other is the self-designed aviation network transmission model. The aggregation between models is more inclined to the exchange of some parameters of the two models, which is actually collaborative training. This article does not consider the disconnection of the blockchain network, assuming that it is carried out under the condition that the network is stable.
Point 11:
Which consensus did the author consider? How data is stored in blockchain? What's the structure of blockchain?
Response 11:
The consensus adopts the raft consensus algorithm, and the data is entered into the blockchain system by filling in the data form of the smart contract. The blockchain structure is a traditional chain structure.
Point 12:
Why did the author choose Hyperledger? Won't it be a private network? Who will be the aggregator in the network then?
Response 12:
Fabric provides a trusted authentication mechanism and sets the entry threshold for collaboration, which meets the requirements of the application scenario. It is a collaborative governance through alliances, rather than the proprietary chain of a specific organization, and the aggregation of the network is achieved by designing specific smart contracts.
Point 13:
An analysis of computational complexity and convergence is required.
Response 13:
The original paper of computational complexity has been analyzed by experiments, and the convergence of the model is also verified by experimental data to fit the actual curve.
Point 14:
Did the authors consider the privacy leakage of the model itself?
Response 14:
Because the federated learning model designed in this paper does not need to pass the entire model, only part of the parameters need to be passed, so the entire model will not be leaked.
Point 15:
A Comparison with existing works is missing. Which DL did the author consider?
Response 15:
The combination of traditional blockchain and federated learning is mostly collaborative training between the same model, and this paper tends to study collaborative training between different models.
Point 16:
Hight quality figures are required for Fig. 3 and 4.
Response 16:
Figures have been replaced as required.
Reviewer 3 Report
How can author conclude this? “However, at present, most national medical centers are often unable to know the flight status of confirmed patients, and most airline companies cannot know the illness status of passengers due to different national policies”.
Problem statement is not properly outlined in section 1.
Authors must enumerate the contributions properly in introduction section.
Advocate the need for aggregator in traditional federated learning systems.
Related work is below par and must be modified by adding more recent works in the area.
Concept of blockchain is not properly presented in the manuscript. Authors must discuss the blockchain concepts in detail. The following works may be useful. Unification of Blockchain and Internet of Things (BIoT): Requirements, working model, challenges and future directions; Untangling blockchain technology: A survey on state of the art, security threats, privacy services, applications and future research directions.
Authors must discuss the obtained results in more detail. How is the proposed scheme better than the existing technique.
The manuscript is low in citing recent references. Authors must cite the recent works in the revised version to enhance the overall readability of the manuscript.
How can author conclude this? “However, at present, most national medical centers are often unable to know the flight status of confirmed patients, and most airline companies cannot know the illness status of passengers due to different national policies”.
Problem statement is not properly outlined in section 1.
Authors must enumerate the contributions properly in introduction section.
Advocate the need for aggregator in traditional federated learning systems.
Related work is below par and must be modified by adding more recent works in the area.
Concept of blockchain is not properly presented in the manuscript. Authors must discuss the blockchain concepts in detail. The following works may be useful. Unification of Blockchain and Internet of Things (BIoT): Requirements, working model, challenges and future directions; Untangling blockchain technology: A survey on state of the art, security threats, privacy services, applications and future research directions.
Authors must discuss the obtained results in more detail. How is the proposed scheme better than the existing technique.
The manuscript is low in citing recent references. Authors must cite the recent works in the revised version to enhance the overall readability of the manuscript.
Author Response
Response to Reviewer 3 Comments
Point 1:
How can author conclude this? “However, at present, most national medical centers are often unable to know the flight status of confirmed patients, and most airline companies cannot know the illness status of passengers due to different national policies”.
Response 1:
We have updated our paper and added support data to this issue.
Point 2:
Problem statement is not properly outlined in section 1.
Response 2:
We have restated section 1 as required.
Point 3:
Authors must enumerate the contributions properly in introduction section.
Response 3:
We have clarified our contributions at the end of the introduction.
Point 4:
Advocate the need for aggregator in traditional federated learning systems.
Response 4:
This article is different from traditional federated learning in that you do not need to pass the entire model, only some parameters. Collaborative learning of two different domain models without sharing their respective training data is supported. Blockchain provides the building blocks for trusted collaboration and secures training.
Point 5:
Related work is below par and must be modified by adding more recent works in the area.
Response 5:
We have reinforced our related works as required. With the help of these works we are enabled to illustrate our idea more clearly.
Point 6:
Concept of blockchain is not properly presented in the manuscript. Authors must discuss the blockchain concepts in detail. The following works may be useful. Unification of Blockchain and Internet of Things (BIoT): Requirements, working model, challenges and future directions; Untangling blockchain technology: A survey on state of the art, security threats, privacy services, applications and future research directions.
Response 6:
We have clarified the concept of blockchain as required.
Point 7:
Authors must discuss the obtained results in more detail. How is the proposed scheme better than the existing technique.
Response 7:
The combination of traditional blockchain and federated learning is mostly collaborative training between the same model, and this paper tends to study collaborative training between different models.
Point 8:
The manuscript is low in citing recent references. Authors must cite the recent works in the revised version to enhance the overall readability of the manuscript.
Response 8:
We have taken more advanced methods into consideration as required and enhanced the overall readability of our work.
Round 2
Reviewer 1 Report
No further comments.
No further comments.
Reviewer 2 Report
I have no further concerns.
I have no further concerns.